REGISTERED REPORT PROTOCOL

# eNose-TB: A trial study protocol of electronic nose for tuberculosis screening in Indonesia

**Antonia Morita Iswari Saktiawati**[1], **Kuwat Triyana**[2], **Siska Dian Wahyuningtias**[1], **Bintari Dwihardiani**[1], **Trisna Julian**[2], **Shidiq Nur Hidayat**[2], **Riris Andono Ahmad**[1,3], **Ari Probandari**[1,4], **Yodi Mahendradhata**[1,5]*

1 Faculty of Medicine, Public Health and Nursing, Center for Tropical Medicine, Universitas Gadjah Mada, Yogyakarta, Indonesia, 2 Faculty of Mathematics and Natural Sciences, Department of Physics, Universitas Gadjah Mada, Yogyakarta, Indonesia, 3 Faculty of Medicine, Public Health and Nursing, Department of Biostatistics, Epidemiology, and Population Health, Universitas Gadjah Mada, Yogyakarta, Indonesia, 4 Faculty of Medicine, Department of Public Health, Universitas Sebelas Maret, Surakarta, Indonesia, 5 Faculty of Medicine, Public Health and Nursing, Department of Health Policy and Management, Universitas Gadjah Mada, Yogyakarta, Indonesia

* ymahendradhata@ugm.ac.id

## Abstract

### Background

Even though conceptually, Tuberculosis (TB) is almost always curable, it is currently the world's leading infectious killer. Patients with pulmonary TB are the source of transmission. Approximately 23% of the world's population is believed to be latently infected with TB bacteria, and 5–15% of them will progress at any point in time to develop the disease. There was a global diagnostic gap of 2.9 million between notifications of new cases and the estimated number of incident cases, and Indonesia carries the third-highest of this gap. Therefore, screening TB among the community is of great importance to prevent further transmission and infection. The electronic nose for screening TB (eNose-TB) project is initiated in Yogyakarta, Indonesia, to screen TB by breath test with an electronic-nose that is easy-to-use, point-of-care, does not expose patients to radiation, and can be produced at low cost.

### Methods/Design

The objectives of the two-phase planned project are to: 1) investigate the potential of an eNose-TB as a screening tool in Indonesia, in comparison with screening with clinical symptoms and chest radiology, which are currently used as a standard, and 2) analyze the time and cost of a screening algorithm with eNose-TB to obtain additional case detection. A cross-sectional study will be conducted in the first phase to validate the eNose-TB. The validation phase will involve 395 presumptive TB patients in the Surakarta General Hospital, Central Java. In the second phase, a cross-sectional research will be conducted, involving 1,383 adults and children in the municipality of Yogyakarta and Kulon Progo district of Yogyakarta Province.

**Data Availability Statement:** All relevant data from this study will be made available upon study completion. This is a registered protocol, thus we can only start data collection when the manuscript has been published and thus we do not have yet the minimum dataset to be uploaded.

**Funding:** The eNose-TB study is supported by the Ministry of Research, Technology and Higher Education of the Republic of Indonesia through the World Class Research grant. The Ministry has no role in the study design, data collection, analysis, decision to publish, or manuscript preparation.

**Competing interests:** The authors have declared that no competing interests exist.

## Discussion

The findings will provide data concerning the sensitivity and specificity of the eNose-TB as a screening tool for tuberculosis, and the time and cost analysis of a screening algorithm with the eNose.

## Trial registration

NCT04567498; https://clinicaltrials.gov/.

## Introduction

Tuberculosis (TB) remains a major global public health problem. TB is an infectious disease that causes the highest number of deaths globally due to a single infectious agent [1]. Indonesia ranks as the second-highest TB-burdened country in the world and carries the third-highest gap between the estimated number of incident cases and the notifications of new cases [1]. Approximately 30% of active TB cases are currently undetected by Indonesia's health services [2].

The World Health Organization (WHO) recommends screening for early detection of tuberculosis cases to reduce transmission and improve patient outcomes [3]. The screening is prioritized for people who have close contacts with TB (who live in the same household or frequent contact with the sputum smear-positive TB patients), and afterward for populations at risk of TB (people with HIV, diabetes, residents of the area with high TB transmission, *i.e.*, slum areas) [3]. Screening with clinical symptoms has a sensitivity of 70% while screening with chest radiography has a sensitivity of 87% (3). However, chest radiography exposes patients to radiation and is not practical to use in remote areas. Due to TB's significant clinical and economic impact, it is of great importance to develop screening tools that are accurate, point-of-care, easy to use, and produced at a low cost, thus can be widely used in lower-middle-income countries such as Indonesia.

An electronic nose (eNose) was investigated as a diagnostic tool for TB by examining the patients' exhaled breath [4,5]. An eNose is an easy-to-use tool based on an array of sensors that can learn and diagnose a disease from the pattern of volatile organic compounds (VOCs) that are produced by *Mycobacterium tuberculosis* [6–8] or by the host metabolism due to TB, which are different from standard conditions [9]. The sensor array in the electronic nose comprises non-specific sensors exposed to a variety of odor-sensitive biological materials, such as breath, urine, or feces. An odor stimulus produces a specific fingerprint from the sensor array. Patterns or fingerprints from known odors are used to build a database and train a pattern recognition system; thus, unknown odors can subsequently be classified. Generally, eNose instruments consist of hardware components to collect and transport odors to the sensor array and electronic circuitry to digitize and store the sensor's responses for signal processing [10]. An eNose is practical to carry, easy to use, and without radiation exposure, making it suitable as a screening tool.

To our knowledge, there has been no published study protocol concerning eNose as a screening tool for TB. To date, trials that investigate eNose as a screening tool for TB are planned or ongoing, which include our trial and another one in Paraguay (NCT04407325). We aim to investigate the potential of an eNose-TB as a screening tool in Indonesia compared to the screening with clinical symptoms and chest radiology, which are currently used as a

standard. We further aim to analyze the time and cost of a screening algorithm with eNose-TB to obtain additional detection of one TB case.

We formulated the following hypotheses: 1) Breath test has high sensitivity (>90%) as a screening test for TB, and 2) the time and cost of a screening algorithm with eNose-TB to obtain additional detection of one TB case are shorter and lower, respectively, in comparison with the standard of screening by clinical symptoms and chest radiology that is currently used.

## Materials and methods

### Design plan

In a previous pilot machine learning study (unpublished), we trained the eNose-TB to recognize the breath pattern of TB patients and healthy controls. The TB patients were recruited consecutively from the Respira Lung Hospital, a hospital formed by five previous public lung clinics in Yogyakarta. Meanwhile, the healthy controls were recruited from TB patients' neighborhood to represent the same socioeconomic conditions, but these participants did not have close contact with the TB patients. The inclusion criteria of the TB patients were: 1) diagnosed as TB, had at least 1+ on the WHO scale in minimum 1 out of 2 samples examined for microscopy, and positive result of Xpert MTB/Rif and culture (identification of *M. tuberculosis*) from the samples, 2) had never received TB treatment, 3) agreed to participate in the study by signing the informed consent, 4) did not smoke for at least one year before the study. The inclusion criteria for the healthy subjects were: 1) agreed to participate in the study by signing the informed consent, 2) did not smoke for at least one year before the study, and 3) had no sign nor symptoms of TB. The exclusion criteria were 1) unable to breathe normally for two minutes due to respiratory illness, 2) incomplete supporting examination data, and 3) invalid measurements of breath tests.

As part of the routine examination, all participants were characterized by clinical symptoms (*e.g.*, persistent cough, unintentional ≥5% weight loss, and night sweats), CXR, smear microscopic, Xpert MTB/Rif examination. For study purposes, we added culture, HIV test, and eNose-TB breath test. The participants breathed normally through a disposable mouthpiece of pipe connected to the electronic nose. The inlet was covered with HEPA-filter to protect the *electronic-nose* contaminated with bacteria and virus. Afterward, the electronic-nose was connected to a laptop, and the breath data was analyzed.

We included 27 TB patients and 24 healthy controls. Bruins *et al.* and Zetola *et al.* [11,12] showed that this was an acceptable sample size to recognize the breathing pattern by the eNose-TB machine. The eNose-TB raw data were manipulated by the baseline using a different method. The data were cleaned by windowing the data during the sampling phase with five windows, and a feature extraction method was performed in each window, namely the maximum value, the median value, and the standard deviation value. Afterward, a leave-one-out cross-validation (LOOCV) was used in training procedures as a validation method. LOOCV was used in the hyper-parameters tuning process of Support Vector Machine (SVM) and Recursive Feature Elimination (RFE) for a feature selection method. Logistic regression was used as an estimator in the RFE method.

Seven breath samples were invalid, thus were excluded from the analysis. Fig 1 shows the Receiver Operating Characteristic (ROC) curve of the best model in sensitivity and specificity of breath test in the training phase; sensitivity was 95% (95% CI = 77–100%), and the specificity was 82% (95% CI = 60–95%) (Table 1). During the breath sample collection, there were no adverse events (e.g., breathless, infection, or bleeding) associated with the study intervention.

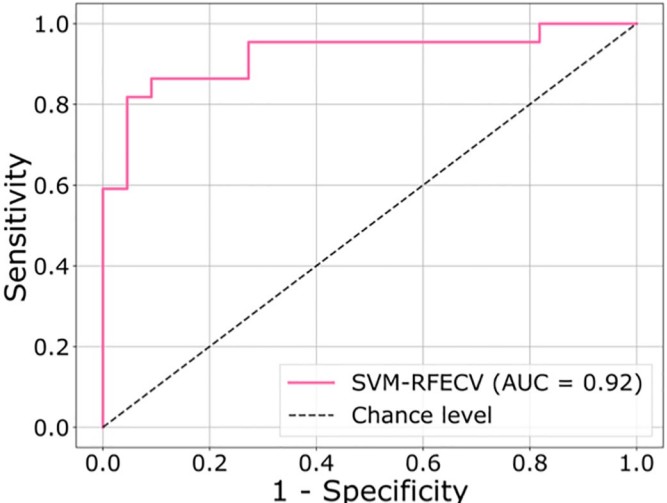

**Fig 1. ROC curve of the best model in diagnostic sensitivity and specificity of breath test in the training phase.**
ROC = Receiver Operating Characteristic.

After the previous pilot machine learning study was finished, we aim to continue the study with a two-phase planned project, namely the validation and screening phases. The purpose of this current study is to validate the eNose-TB and investigate it as a screening tool for TB.

**I. Validation phase of the electronic nose.** This phase uses a cross-sectional design. It is conducted in Surakarta General Hospital, Central Java, and primary health centers in the municipality of Yogyakarta and Kulon Progo district. The inclusion criteria are: 1) agree to participate in the study, 2) able to produce samples for Xpert MTB/Rif examination, and 3) able to produce exhaled air samples. The exclusion criteria are: 1) invalid measurements of breath tests, 2) incomplete CXR data, 3) missing specimens, and 4) inability to breathe normally for 2 minutes due to respiratory illness. All study participants are characterized by clinical symptoms (*e.g.*, persistent cough, unintentional ≥5% weight loss, night sweats), CXR, smear microscopic, Xpert MTB/Rif examination, HIV testing, and breath test with the eNose-TB (Fig 2). Participants' sputum is taken and analyzed with Xpert MTB/Rif examinations to confirm the diagnosis of TB. An acid-fast-bacilli Ziehl-Neelsen microscopy is done as a part of the routine examination in the hospital. If a participant cannot produce sputum, he/she will be asked to collect a stool sample, as this method has been shown as reliable in diagnosing TB [13].

The study uses triple-blind masking, in which the research subjects, breath sample takers, and laboratory sample examiners do not know the results of each sampling that has been done. The final data processor is also blinded to the results of Xpert or smear microscopy. The

**Table 1. Performance of breath test results in the training phase.**

|  | Final diagnosis | | Sensitivity (%) (95% CI) | Specificity (%) (95% CI) |
|---|---|---|---|---|
|  | PTB (n = 22) | No PTB (n = 22) |  |  |
| Positive breath test | 21 | 4 | 95 (77–100) | 82 (60–95) |
| Negative breath test | 1 | 18 |  |  |

PTB: Pulmonary tuberculosis, AUC: Area under the curve, CI: Confidence interval.
[a]generated from the real value.

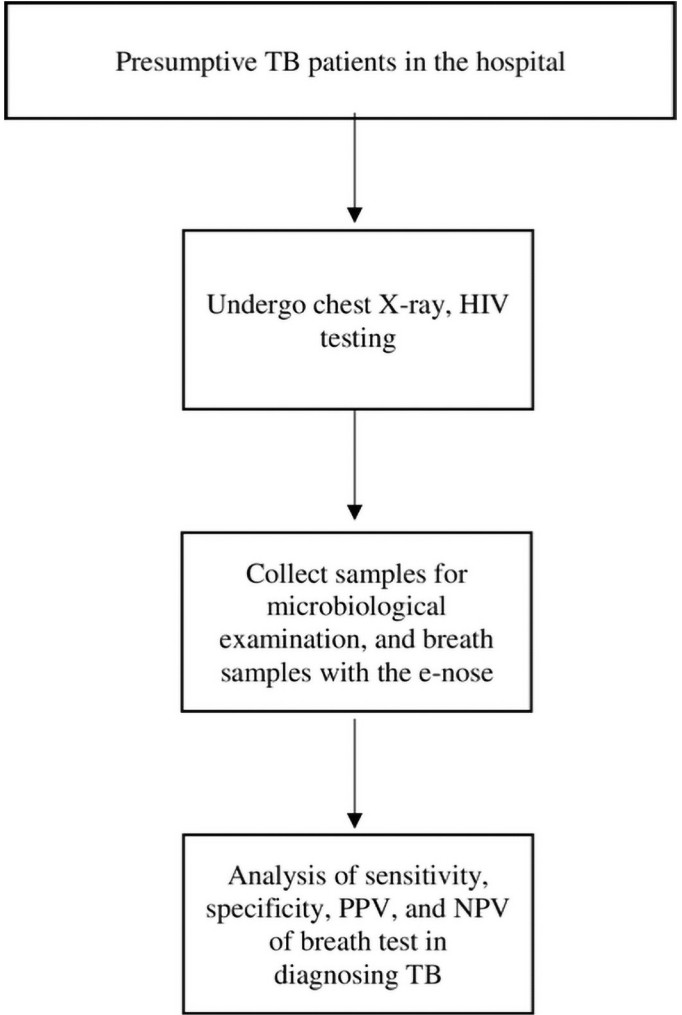

**Fig 2. Flowchart of the validation phase of the eNose-TB in diagnosing TB.**

breath sampling data are saved in graphic form, of which interpretation will be done later by the data processor at the final stage.

**II. Screening phase of the electronic nose.** After the validation phase, we will conduct a screening phase with a cross-sectional design. The study will be conducted in the municipality of Yogyakarta, a region with the highest TB prevalence in Yogyakarta Special Province, and in the Kulon Progo district of Yogyakarta, which contains many remote areas, where eNose-TB will have its most useful application. The municipality of Yogyakarta has 18 primary health centers and 21 hospitals [14]. The estimated TB incidence in 2019 is 1,400 cases [15]. The Kulon Progo district of Yogyakarta has 21 primary health centers and nine hospitals [16], and the estimated TB incidence in 2019 is 1,033 cases [15]. The study population is adults and children in these two districts. In this phase, we will use an eNose-TB that has been validated; thus, it has a more robust trained pattern recognition technique, and as a result, it can give real-time measurements.

As the active TB case finding activity, namely "Zero TB city", is launched in Yogyakarta, the eNose-TB will be paired with the symptom screening and chest radiology (CXR) in this activity. The mobile clinic team consisting of doctors, radiology officers, laboratory personnel, and

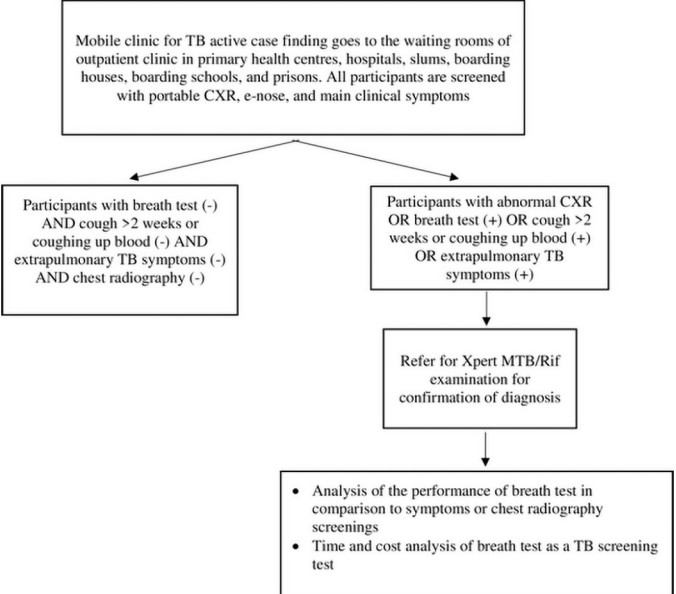

**Fig 3. Flowchart of screening with the eNose-TB, clinical symptoms, and chest radiography.**

nurses will travel to the area with a high risk of TB, *i.e.*, the waiting rooms of an outpatient clinic in primary health centers and hospitals, slums, boarding houses, boarding schools, and prisons. The inclusion criteria are: 1) agree to participate in the study, and 2) currently not in TB treatment, while the exclusion criteria are: 1) invalid measurements of breath tests, 2) incomplete CXR data, 3) missing specimens, and 4) unable to breathe normally for 2 minutes due to respiratory illness.

Participants will be interviewed for symptoms of TB, which consist of the main symptoms (cough that lasts more than two weeks, night sweats, unintentional ≥5% weight loss) and other symptoms (coughing up blood, fever >1 month, enlarged lymph nodes, shortness of breath and chest pain), and undergo CXR examination. They will then undergo a breath test. Patients with positive radiological results or positive breath tests or coughing for more than two weeks or coughing up blood or showing extrapulmonary TB symptoms will be asked to collect their sputum or stool samples for Xpert MTB/Rif examinations to confirm the diagnosis of TB (Fig 3).

We will collect participants' demographic and clinical data. We will analyze the screening algorithm's time and cost with eNose-TB to obtain additional detection of 1 TB case. The screening phase also uses triple-blind masking, in which the research subjects, breath sample takers, and laboratory sample examiners do not know the results of each sampling.

## Tools and materials

Figs 4 and 5 show the eNose-TB system, which contains a sampling system (air collecting bag, HEPA-filter, valve, and micro-pump), a sensor array system, and a data acquisition system (Arduino and a personal computer). The micro-pump has a flow rate of about 1 milliliter per minute. This micro-pump functions to suck the breath sample into the gas sensor chamber and to push it out after the sensing process is complete. Meanwhile, HEPA filters are used to filter virus particles or other foreign objects other than VOCs from breath samples. The manufacturer of the eNose-TB device is the Department of Physics, Faculty of Mathematics and

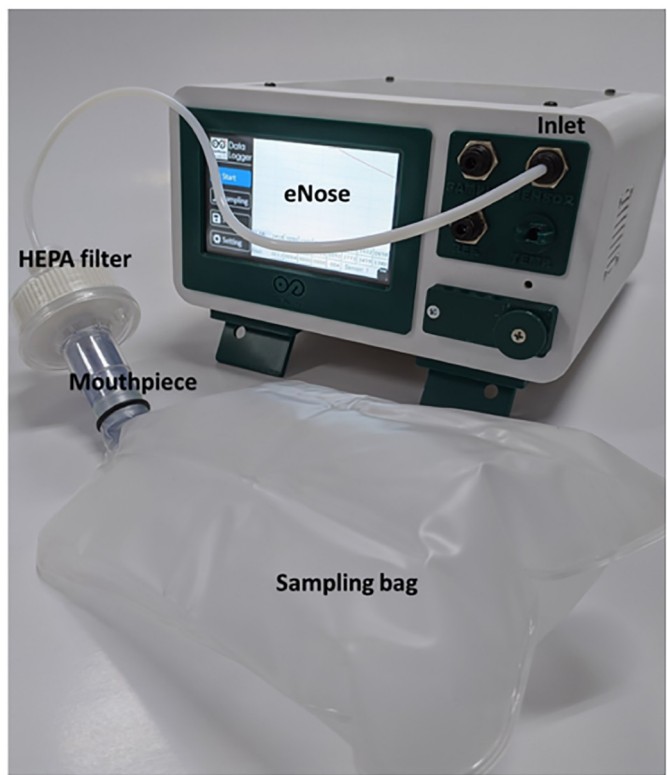

**Fig 4. The lab-made eNose-TB device connected to the sampling bag through a mouthpiece and HEPA filter.**

Natural Sciences, Universitas Gadjah Mada, Indonesia. The software was also developed and validated by the same department.

We use the Metal-Oxide-Semiconductor (MOS)-based gas sensors. All sensors used in the eNose-TB device are commercial sensors available in the market with a high level of replication [17]. There are 16 sensors used, with the characteristics of global selectivity. Based on the datasheet and our tests' results, each sensor responds to at least 5 different gases, thus the detection range is at least 80 different VOC groups. With that coverage, the sensor array will form a relatively consistent pattern for each group of breath.

Fig 6 is a typical voltage response of 16 sensors in eNose-TB device to time. The sensor response begins with the baseline at the same level for 10 seconds, when the surrounding air is introduced into the gas sensor room. After that, the electronic valve will control that the breath sample is inserted into the gas sensor chamber for 120 s. The sensing process occurs after the 10th seconds to the 130th seconds, so the total sampling time is 120 seconds. When the sensing process is complete, the sensor room is cleaned with the ambient air until it returns to the

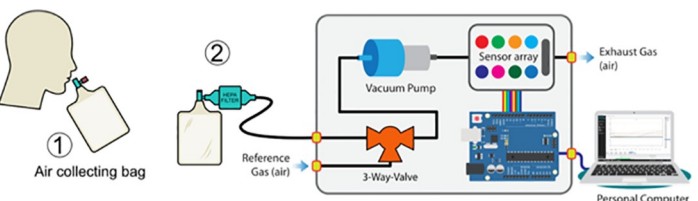

**Fig 5. Schematic circuit of the eNose-TB system.**

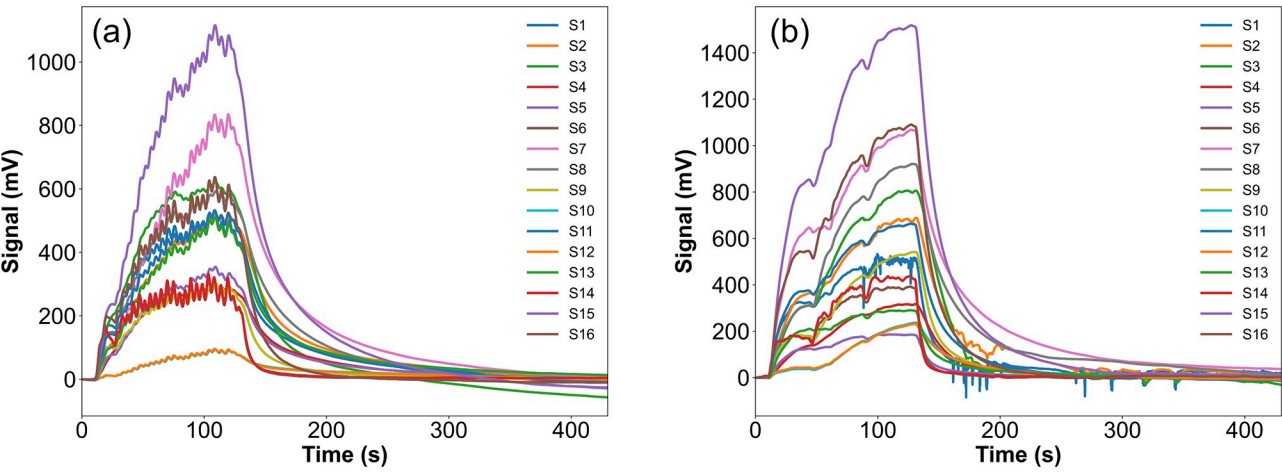

**Fig 6. The response of 16 sensors of eNose-TB for negative TB (a) and positive TB (b) subjects.**

original baseline level. To distinguish the breath response pattern between the negative and positive TB subjects, each sensor's response during the sensing process is divided into 5 windows with an interval of 24 seconds each.

Furthermore, we extract the features from each window in the form of the median and standard deviation values with pre-processing in the form of standard and scaler. Each sensor has 10 characteristic values, which are then used as input into the machine learning model as a classification model, namely the support vector machine (SVM). The three main parameters of SVM for each set is a cost value of 10, with the kernel in the form of a linear function and using an optimizer in the form of a recursive feature elimination (RFE). Because the number of data in the pilot machine learning phase (training phase) was limited, we used a system performance validation method in the form of a leave-one-out cross-validation (LOO-CV).

The effects of temperature and humidity in the sensor room are measured in parallel (with a calibrated SHT31 sensor) during the sensing process. Fig 7 shows the change in temperature and relative humidity during the pilot machine learning study (training phase). Changes in

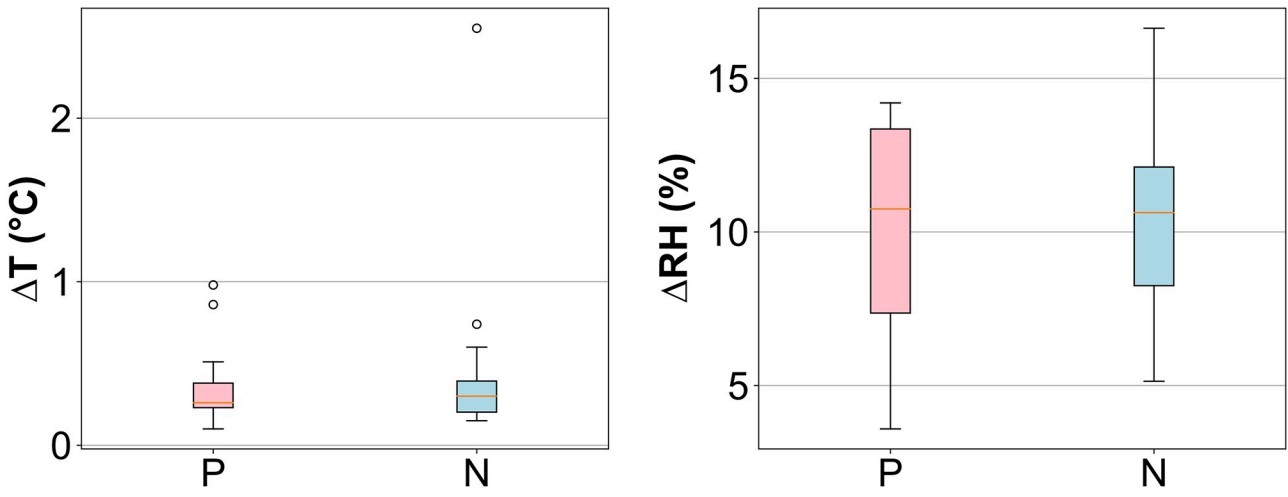

**Fig 7. Change of temperature (a) and relative humidity (b) in the sensor chamber during the measurement.**

temperature and relative humidity during measurement were only about 0.5 C and 15%, respectively, which means that there is no significant effect on sensor response when referring to sensor characteristics in general [17].

**eNose-TB software system.** The eNose-TB software system consists of two programming parts: microcontroller programming (Arduino MEGA 2560) and data logger software programming (Windows). Microcontroller programming is conducted with the Arduino IDE software, while the data logger programming uses Microsoft Visual Studio 2019 Community (open source) software, which uses the C# programming language.

## Sampling plan

**Collection of the exhaled breath samples.** The study participants are asked to take two initial deep breaths and exhale, using their protective masks. Next, subjects are asked to take a deep breath and exhale powerfully (a forced expiratory volume) to the air collecting bag from the third and subsequent breaths (maximum until the fifth breath) until the collecting bag is full. The collecting bag is sealed and connected to the eNose-TB machine via a collecting hose and HEPA-filter protecting the eNose-TB from microbes. A designated officer supervises the breath collection. The data are read and stored in the eNose-TB machine. Considering the risks of TB and COVID-19 transmission, we will place the eNose-TB device in a particular isolation room, and the officer who operates the eNose-TB uses level 3 personal protective equipment (N95 mask, gloves, goggles, gown, boots, or closed shoes). We also use disposable equipment (HEPA-filter, connector, and air collecting bag), apply disinfectant procedure after each breath sampling, and ensure that the physical distancing measurement is applied during the sampling collection. We plan to finish the data collection in December 2021, with the possibility of extending our study to another year and the addition of study sites if COVID-19 jeopardizes our plans.

**Variables.** The outcome variable in the first phase (validation phase) will be sensitivity, specificity, positive predictive value (PPV), and negative predictive value (NPV), while in the second phase (screening phase) will be positive agreement, negative agreement, and time and cost analysis of screening with the eNose-TB. We will also collect the following data variable: age, body mass index, sex, occupation, smoking habits, alcohol consumption, comorbidities (diabetes, asthma, Chronic Obstructive Pulmonary Disease/COPD, HIV/AIDS, flu, bronchitis, bronchiectasis, lung fibrosis, lung abscess, empyema, polycystic lung disease), co-medication (inhalation drugs and antibiotics), and food and beverages consumption before the breath test. Data of age, body mass index, sex, comorbidities, co-medication, and occupation will be asked directly to patients or their legal guardians. Smoking habits and alcohol consumption will be measured by asking participants, "When was the last time you smoked" and "When was the last time you consumed alcohol" (on a value of 1 and 2, 1 being 'never or $\geq$8 hours ago', 2 being '<8 hours ago'). Food and beverage consumption before the breath test will be measured by asking participants, "When did you last take food or beverages" (on a value of 1 and 2, 1 being 'not having prior intake or having prior intake $\geq$1 hour ago', 2 being 'having prior intake <1 hour ago'), and "What kind of food and drink did you consume" (will be noted as description data, and later categorized as poultry meat, coffee, tea, milk, and none of them). All questions will be asked in the local language (Bahasa Indonesia). The interview guide can be accessed in the S1 Appendix. Two researchers will double-enter all data into a database and ensure no missing data or typing errors.

**Sample size.** The validation phase is conducted on minimum 395 presumptive TB patients who are recruited consecutively. The sample size is calculated based on the following forma: $n = \frac{Z_{\propto/2}{}^2 x SN x (1-SN)}{d^2}/P$ [18] with $Z_{\propto/2}$ is the Z-score for the confidence level of 95% (1.96),

SN is the pre-determined value of sensitivity (90%), $d$ is the maximum marginal error (5%), and $P$ is the disease prevalence in the study site (35%). Based on the proportion of TB cases in children among all cases in Indonesia, which is approximately 10% [19], we will recruit 40 children among all participants. The lower limit in age for child-participants is 4 years old, as they are considered able to follow the instructions for breath collection. In case the participants cannot produce sputum for the Xpert MTB/Rif examination, they will be asked to collect their stool samples, as this method has been shown as reliable in diagnosing TB [13].

Based on the same formula as above [18], with $Z_{\propto/2}$ value of 1.96, SN value of 90%, d value of 5%, and $P$ value of 10%, the minimum number of study participants needed in the screening phase is 1383 patients. We will recruit the participants consecutively. Based on the proportion of TB cases in children [19], the number of children recruited is 140 among all participants.

## Analysis plan

**Microbiological analysis.** The samples that are taken will be analyzed with acid-fast-bacilli Ziehl-Neelsen microscopy and Xpert MTB/Rif, which are conducted according to the WHO [20] and manufacturer guidelines, respectively, in the study sites' laboratories. When the facilities are unavailable, the samples are referred to the TB-Microbiology laboratory, Faculty of Medicine, Public Health, and Nursing, Universitas Gadjah Mada.

**Data interpretation of eNose-TB.** Multivariate (chemometric) data analysis uses the open-source programming language R version 3.5.1 and Python version 3.7. The feature extraction method is conducted by taking the average value of each sensor's sampling data.

Evaluation of all datasets using SVM model resulted in the confusion matrix. The data is divided into two classes, *i.e.*, the class predicting the classifier's results and the actual class. In measuring performance using a confusion matrix, four conditions represent the results of the classification process, namely true positive (TP), true negative (TN), false positive (FP), and false-negative (FN). A ROC curve is used to perform analysis of 2 label classes. The caret library is also used in the ROC analysis procedure.

**Statistical analysis.** In the first phase (validation phase), we will calculate the sensitivity, specificity, positive predictive value (PPV), and negative predictive value (NPV) of the breath test using the Xpert MTB/Rif as the reference standard. In each variable (such as age, body mass index), one stratum's ROC-curve indicating the breath test's sensitivity and specificity is compared with another stratum's ROC-curve. An association between the breath test's variable and sensitivity-specificity is indicated by a significant difference of an AUC between strata ($p < 0.05$). In the second phase (screening phase), the performance of screening with a breath test will be compared with screening with clinical symptoms or CXR examination by calculating positive and negative agreements between the breath test and clinical symptoms or CXR examination. The time and cost of a screening algorithm with eNose-TB to obtain additional detection of one TB case will be calculated as the mean of time needed and mean of cost spent from the beginning of screening with the eNose-TB until the detection of the case. Statistical analysis is performed using STATA/SE 15 (License: Universitas Gadjah Mada).

## Ethical considerations

The research will be conducted following the principles of the Helsinki Declaration of 2013 and Good Clinical Practice [21]. The research has obtained the approval of the institutional review board at the Faculty of Medicine, Public Health, and Nursing, Universitas Gadjah Mada, Yogyakarta, Indonesia (KE/FK/1127/EC/2020) and has been registered in clinicaltrials. gov (NCT04567498). All subjects will sign an informed consent before participating in this study.

In the validation phase, the attending physician at the study sites will provide information to patients regarding this study and offer patients a chance to meet one of the research teams or staff at the hospital trained to provide research information. The research team or staff provides information about this research verbally, based on what is written in the informed consent form. In the screening phase, nurses in the mobile clinic team will provide information to prospective participants about this study. All participants need to sign the informed consent form to participate in this research, and they are given 24 hours to decide to participate in this study. Subjects can leave the study at any time for any reason if they wish to do so without any consequences. The investigator can decide to withdraw a subject from the study for urgent medical reasons. If they withdraw, they will be diagnosed and treated with the standard treatment according to local guidelines.

The breath sampling is conducted for a maximum of three breaths consecutively to avoid a hyperventilation-induced response when someone deep inhales and exhales 6–10 breaths consecutively. If a hyperventilation-induced response happens, the health officer will ask the participants to have rest by sitting or lying down for a while. If the subjects suffer from any adverse event, the investigator will treat them until they recover. Participants will not be paid to part in this research, but if there are costs incurred by participants to participate in this study or due to any adverse events, these costs will be compensated.

The investigator will submit a summary of the study progress to the accredited institutional review board once a year and submit a protocol amendment immediately when the investigator makes one. Information will be provided on the date of inclusion of the first subject, numbers of subjects included and numbers of subjects that have completed the study, serious adverse events/serious adverse reactions, other problems, and amendments.

The research team will store all data in a safe, locked cabinet, and only the research team has access to this data, or legal authorities and audit officers. The legal authorities and audit officers who are independent of the sponsor have access to these data and its analysis and can decide to terminate the study. The audit is conducted once a year by visiting the study site and examining the study data. Physicians who are not involved in the study (independent physicians) are also provided access to answer the prospective participants' questions. The data will be made available upon study completion in keeping with the PLOS Data policy. The investigators will communicate the study results via publication and report to the sponsor and study sites.

## Differences from the study protocol

Indonesia was written as the country with the third-largest burden of TB in the study protocol, as the 2020 edition of the WHO global TB report was not yet issued at the time of study protocol submission to the institutional review board. Meanwhile, when we submitted this manuscript, we used the latest data from the 2020 edition of the WHO global TB report. In this manuscript, we included the latest figure of the schematic circuit of the eNose-TB system, while the study protocol still used the previous figure of the eNose-TB system. In this manuscript, we wrote the number of child participants that will be recruited and added information regarding collecting and managing reported adverse events and other unintended effects of study interventions, plan of an audit of the study conduct, and the dissemination policy.

## Discussion

This is the first study protocol investigating the potency of an eNose-TB as a screening tool for TB. Based on this screening study's findings, the extent of the potential of eNose-TB as a screening tool for TB, and the time and cost analysis of a screening algorithm with an eNose-

TB would be known. The findings will enable the stakeholders and health professionals to implement effective screening strategies. The study is conducted during the time of the COVID-19 pandemic; thus, additional precautions are taken, such as using the appropriate protective personal equipment in collecting the samples, placing the eNose-TB device in a particular isolation room, using disposable equipment (HEPA-filter, connector, and air collecting bag), applying disinfectant procedure after each breath sampling, and ensuring that the physical distancing measurement is applied during the sampling collection.

## Supporting information

**S1 File. SIPIRIT checklist.**
(DOC)

**S2 File. Study protocol in Indonesian language.**
(DOCX)

**S3 File. Study protocol in English.**
(DOCX)

**S1 Appendix. Interview guide.**
(PDF)

## Acknowledgments

We would like to thank the staff at Surakarta General Hospital and the primary health centers for their cooperation for the upcoming study.

## Author Contributions

**Conceptualization:** Antonia Morita Iswari Saktiawati, Kuwat Triyana, Bintari Dwihardiani, Riris Andono Ahmad, Ari Probandari, Yodi Mahendradhata.

**Data curation:** Antonia Morita Iswari Saktiawati, Kuwat Triyana, Trisna Julian, Shidiq Nur Hidayat.

**Formal analysis:** Antonia Morita Iswari Saktiawati, Kuwat Triyana, Bintari Dwihardiani, Trisna Julian, Riris Andono Ahmad, Ari Probandari, Yodi Mahendradhata.

**Funding acquisition:** Riris Andono Ahmad, Ari Probandari, Yodi Mahendradhata.

**Investigation:** Antonia Morita Iswari Saktiawati, Siska Dian Wahyuningtias.

**Methodology:** Antonia Morita Iswari Saktiawati, Kuwat Triyana, Bintari Dwihardiani, Riris Andono Ahmad, Ari Probandari, Yodi Mahendradhata.

**Project administration:** Antonia Morita Iswari Saktiawati, Siska Dian Wahyuningtias, Yodi Mahendradhata.

**Resources:** Kuwat Triyana, Bintari Dwihardiani, Trisna Julian, Shidiq Nur Hidayat, Ari Probandari, Yodi Mahendradhata.

**Software:** Kuwat Triyana, Shidiq Nur Hidayat.

**Supervision:** Yodi Mahendradhata.

**Validation:** Bintari Dwihardiani, Riris Andono Ahmad, Ari Probandari.

**Visualization:** Antonia Morita Iswari Saktiawati, Kuwat Triyana, Siska Dian Wahyuningtias.

**Writing – original draft:** Antonia Morita Iswari Saktiawati.

**Writing – review & editing:** Antonia Morita Iswari Saktiawati, Kuwat Triyana, Siska Dian Wahyuningtias, Bintari Dwihardiani, Trisna Julian, Shidiq Nur Hidayat, Riris Andono Ahmad, Ari Probandari, Yodi Mahendradhata.

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
