## [Decision Letter · Decision Letter 0]

22 Jan 2021

PONE-D-20-33305

eNose-TB: a trial study protocol of electronic nose for tuberculosis screening in Indonesia

PLOS ONE

Dear Dr. Mahendradhata,

Thank you for submitting your manuscript to PLOS ONE. After careful consideration, we feel that it has merit but does not fully meet PLOS ONE’s publication criteria as it currently stands. Therefore, we invite you to submit a revised version of the manuscript that addresses the points raised during the review process.

Please submit your revised manuscript. If you will need significantly more time to complete your revisions, please reply to this message or contact the journal office at plosone@plos.org. Please include the following items when submitting your revised manuscript:

We look forward to receiving your revised manuscript.

Kind regards,

Frederick Quinn

Academic Editor

PLOS ONE

Journal Requirements:

3. Please include additional information regarding the interview guide used in the study (lines 259-267) and ensure that you have provided sufficient details that others could replicate the analyses. For instance, if you developed the guide as part of this study and it is not under a copyright more restrictive than CC-BY, please include a copy, in both the original language and English, as Supporting Information.

4.We note that Figure(s) [7, 8 and 9] in your submission contain copyrighted images. All PLOS content is published under the Creative Commons Attribution License (CC BY 4.0), which means that the manuscript, images, and Supporting Information files will be freely available online, and any third party is permitted to access, download, copy, distribute, and use these materials in any way, even commercially, with proper attribution. For more information, see our copyright guidelines: http://journals.plos.org/plosone/s/licenses-and-copyright.

a.         You may seek permission from the original copyright holder of Figure(s) [7, 8 and 9] to publish the content specifically under the CC BY 4.0 license.

Reviewers' comments:

Reviewer's Responses to Questions

**Comments to the Author**

1. Does the manuscript provide a valid rationale for the proposed study, with clearly identified and justified research questions?

Reviewer #1: Yes

Reviewer #2: Yes

2. Is the protocol technically sound and planned in a manner that will lead to a meaningful outcome and allow testing the stated hypotheses?

Reviewer #1: Yes

Reviewer #2: Partly

3. Is the methodology feasible and described in sufficient detail to allow the work to be replicable?

Reviewer #1: Yes

Reviewer #2: Yes

4. Have the authors described where all data underlying the findings will be made available when the study is complete?

Reviewer #1: Yes

Reviewer #2: Yes

5. Is the manuscript presented in an intelligible fashion and written in standard English?

Reviewer #1: Yes

Reviewer #2: No

6. Review Comments to the Author

You may also provide optional suggestions and comments to authors that they might find helpful in planning their study.

Reviewer #1: This is an excellent plan addressing an important question, and making use of a plan in progress to thrive for a zero-TB city of Yogyakarta. The plan is carefully designed, with a pre-specified sample size to develop and validate the test; and next, to sue is for screening for TB.

1) If I understand it correctly, the measurement of breath samples collected will be processed by e-nose on the spot; if that is correct, how will the safety precautions to prevent the transmission of TB and Covid-19 be secured? will all sites have a dedicated room for e-nose measurements?

2) the protocol aims at enrolling 40 children; will there be a loer limit in age, e.g., 10 years and over, to make sure that the child-participants have adult-phenotype pulmonary TB and that these young participants can follow the instructions for breath collection, and produce sputum for GeneXpert?

3) who is the sponsor of the study?

4) who is the manufacturer of the e-nose device, and the software?

5) you plan to complete data collection by December, 2012; what if Covid-19 jeopardizes your plans?

textual suggestions:

page 4, line 82: due to infections from TB: change into 'due to TB'

page 8, line 172: unable to: change into inability to . . 

page 13, line 287: Afterward,  . . change into: Next,  .. 

line 292: As the anticipation of Covid transmission . . . considering the risks of TB and Covid-19 transmission, . . 

page 14, line 303: smoking habits; my impression was that you plan to exclude smokers; please explain.

line 310: 'when was last time you smoked' (I guess you will be asking questions in local languages or Bahasa Indonesia; when was the last time . .(add 'the')

line 313: 'when did you last take food or beaverages'

page 21, line 459; is it a vagal reflex, or perhaps more likely, a hyperventilation-induced response, caused by hypocarbia and respiratory alkalosis? I think that this is what should be anticipated; Trendelenburg positioning is not indicated then, and just a bit of rest, sitting or lying down a while will settle the case. This is much easier than Trendelenburg positioning which will also be more uncomfortable for participants of the study.

Reviewer #2: The objectives are good. TB is a particular problem, especially amongst childrene and it i very difficult to separate from other symptoms to give the right interventions for control strategies. This manuscript is very confusing. It is all written in the present tense. It should be in the past tense as the study, I assume has already been done. It is very difficult to understand the initial study and then the validation study. Were these done in sequence or was the same set of data used for both?? Most of the Figures are totally irrelevant to the objectives. Where are the PCS, where are the clustered dendograms or similar to demonstrate that it is possible to distinguis using their enose the TB vs no TB cohorts and the difference in VOC profiles?? Most of the Figures are about the software system and how the screen pages in the instrument look. These are irrelevant How good is the replication of the sensors used in this instrument? How many sensors are used?? What is the range of detection of different VOC groups. Are they complimentary or what?? None of this information is present. Were sputum samples taken and analysed to confirm TB or no TB patients?? Most of the relevant data around these questions are not included. What about the sensitivity of metal oxide sensors to relative humidity. How was this overcome???

The key scientific aspects are ignored while the process of the software in the instrument are focused on. Unfortunately, this does not help in understanding the Results. This manuscript needs to be completely rewritten in a format which is understandable and approaches the work systematically. I assume the work was all done and needs to be worded in the past tense. The work is interesting and would be useful but the present version is really not up to scratch and needs very major rewriting and presentation.

7. PLOS authors have the option to publish the peer review history of their article (what does this mean?). If published, this will include your full peer review and any attached files.

Reviewer #1: No

Reviewer #2: No

---

## [Author Response · Author response to Decision Letter 0]

8 Mar 2021

We have uploaded a file providing point-by-point response to the comments we've received

---

## [Decision Letter · Decision Letter 1]

23 Mar 2021

eNose-TB: a trial study protocol of electronic nose for tuberculosis screening in Indonesia

PONE-D-20-33305R1

Dear Dr. Mahendradhata,

We’re pleased to inform you that your manuscript has been judged scientifically suitable for publication and will be formally accepted for publication once it meets all outstanding technical requirements.

Kind regards,

Frederick Quinn

Academic Editor

PLOS ONE

Additional Editor Comments (optional):

Reviewers' comments:

Reviewer's Responses to Questions

**Comments to the Author**

1. Does the manuscript provide a valid rationale for the proposed study, with clearly identified and justified research questions?

Reviewer #1: Yes

2. Is the protocol technically sound and planned in a manner that will lead to a meaningful outcome and allow testing the stated hypotheses?

Reviewer #1: Yes

3. Is the methodology feasible and described in sufficient detail to allow the work to be replicable?

Reviewer #1: Yes

4. Have the authors described where all data underlying the findings will be made available when the study is complete?

Reviewer #1: Yes

5. Is the manuscript presented in an intelligible fashion and written in standard English?

Reviewer #1: Yes

6. Review Comments to the Author

You may also provide optional suggestions and comments to authors that they might find helpful in planning their study.

Reviewer #1: All my questions and queries have been appropriately addressed in the revised version; the current revision provides a clear plan for the planned research

7. PLOS authors have the option to publish the peer review history of their article (what does this mean?). If published, this will include your full peer review and any attached files.

Reviewer #1: No

---

## [Editor Report · Acceptance letter]

6 Apr 2021

PONE-D-20-33305R1 

eNose-TB: a trial study protocol of electronic nose for tuberculosis screening in Indonesia 

Dear Dr. Mahendradhata:

I'm pleased to inform you that your manuscript has been deemed suitable for publication in PLOS ONE. Congratulations! Your manuscript is now with our production department. 

Kind regards, 

on behalf of

Dr. Frederick Quinn 

Academic Editor

PLOS ONE